# Pre-RMSNorm and Pre-CRMSNorm Transformers: Equivalent and Efficient Pre-LN Transformers

**Zixuan Jiang, Jiaqi Gu, Hanqing Zhu, David Z. Pan**
Chandra Department of Electrical and Computer Engineering *
The University of Texas at Austin
Austin, Texas, 78712
{zixuan, jqgu, hqzhu}@utexas.edu, dpan@ece.utexas.edu

## Abstract

Transformers have achieved great success in machine learning applications. Normalization techniques, such as Layer Normalization (LayerNorm, LN) and Root Mean Square Normalization (RMSNorm), play a critical role in accelerating and stabilizing the training of Transformers. While LayerNorm recenters and rescales input vectors, RMSNorm only rescales the vectors by their RMS value. Despite being more computationally efficient, RMSNorm may compromise the representation ability of Transformers. There is currently no consensus regarding the preferred normalization technique, as some models employ LayerNorm while others utilize RMSNorm, especially in recent large language models. It is challenging to convert Transformers with one normalization to the other type. While there is an ongoing disagreement between the two normalization types, we propose a solution to **unify** two mainstream Transformer architectures, Pre-LN and Pre-RMSNorm Transformers. By removing the inherent redundant mean information in the main branch of Pre-LN Transformers, we can reduce LayerNorm to RMSNorm, achieving higher efficiency. We further propose the Compressed RMSNorm (CRMSNorm) and Pre-CRMSNorm Transformer based on a lossless compression of the zero-mean vectors. We formally establish the **equivalence** of Pre-LN, Pre-RMSNorm, and Pre-CRMSNorm Transformer variants in both training and inference. It implies that Pre-LN Transformers can be substituted with Pre-(C)RMSNorm counterparts at almost no cost, offering the same arithmetic functionality along with **free efficiency improvement**. Experiments demonstrate that we can reduce the training and inference time of Pre-LN Transformers by $1\% - 10\%$.

## 1   Introduction

Transformers have become a successful architecture for a wide range of machine learning applications, including natural language [39], computer vision [13], and reinforcement learning [7]. It is one of the foundation models [2], and pretrained Transformers [29] demonstrated impressive generalization results. Among the components of Transformers, normalization plays a critical role in accelerating and stabilizing the training process [21]. Layer Normalization (LayerNorm, LN) [1] and Root Mean Square Normalization (RMSNorm) [44] are two common normalization layers in Transformers. LayerNorm is in the original Transformer architecture [39], recentering and rescaling the input vector in $\mathbb{R}^d$ to obtain a zero-mean and unit-variance output. RMSNorm only rescales the input vector with its RMS value, offering greater computational efficiency than LayerNorm.

---

*This work was done when all the authors were at UT Austin. Currently, Zixuan Jiang is at Google, and Jiaqi Gu is at Arizona State University.

37th Conference on Neural Information Processing Systems (NeurIPS 2023).

The machine learning community does not reach a consensus regarding the preferred normalization technique for Transformers. LayerNorm demonstrates remarkable success in the milestone Transformers, such as GPT [30, 5] and ViT [13]. It is still the default normalization layer when building a new Transformer. On the contrary, RMSNorm is reported to accelerate the training and inference with similar performance as LayerNorm in Transformers. It has gained popularity in recent large language models, such as T5 [32], Gopher [31], Chinchilla [16], and LLaMA [38]. However, concerns persist regarding the potential negative impact of RMSNorm on the representation ability of Transformers. It remains an open question to determine the preferred normalization type for Transformers, requiring further theoretical and empirical investigation.

In this work, we aim to mitigate the discrepancy between LayerNorm and RMSNorm in Transformers. When delving into the prevalent Transformer architectures, Pre-LN and Pre-RMSNorm Transformers, we identify an opportunity to **unify** them by removing the inherent redundancy in Pre-LN models. In particular, the main branch vectors in the Pre-LN Transformers are always normalized before they are used, implying that the mean information is redundant. We can recenter the main branch without impact on the functionality of the models, which allows us to reduce LayerNorm to RMSNorm.

We further propose Compressed RMSNorm (CRMSNorm), which takes a vector in $\mathbb{R}^{d-1}$ as input, decompresses it to a zero-mean vector in $\mathbb{R}^d$, and applies the RMSNorm on the decompressed vector. Building upon this new normalization, we propose Pre-CRMSNorm Transformer that employs lossless compression on the zero-mean vectors. We apply such compression to zero-mean activations and parameters in Pre-RMSNorm Transformers, enhancing the efficiency of Pre-RMSNorm Transformers while preserving the equivalent arithmetic functionality.

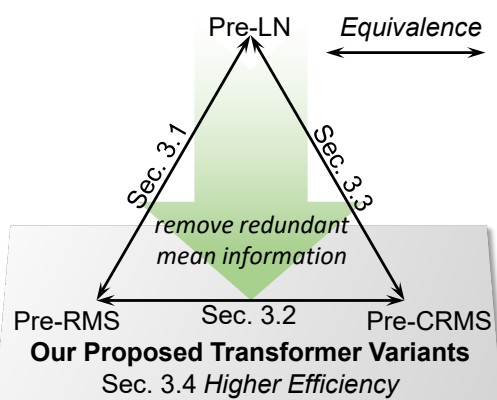

Figure 1: Overview of the three equivalent Transformer variants.

We formally claim that Pre-LN, Pre-RMSNorm, and Pre-CRMSNorm Transformers are equivalent for both training and inference. Figure 1 visualizes the overview of their equivalence. Such equivalence directly enables more efficient training and deployment of Pre-LN Transformers. We can translate a Pre-LN model into an equivalent Pre-(C)RMSNorm model, which can be readily deployed or adapted to downstream tasks. The conversion process incurs minimal costs. We can also train a Pre-(C)RMSNorm model directly as if we train an equivalent Pre-LN Transformer counterpart.

Although our relative improvement in the training and inference efficiency seems not large (up to $10\%$ time reduction), we have the following arguments to support its significance. (1) Our proposed method can guarantee arithmetic equivalence and is a free lunch for Pre-LN Transformers. The efficiency improvement originates from removing the inherent redundancy in Pre-LN Transformers without introducing any fine-tuning or calibration. Our work can strictly push the performance-efficiency **Pareto frontier** of Pre-LN Transformers. (2) The modest relative improvement can be translated into **significant absolute improvement** given that Pre-LN Transformers are foundation models for the current and future data-centric [42] and generative [6] artificial intelligence. For instance, reducing the ChatGPT inference cost by $1\%$ may save $\$7,000$ per day [25]. (3) Our method is **orthogonal and complementary** to most work improving efficiency, such as efficient Transformer variants [35], quantization [22, 3], and distillation to smaller models [34].

We highlight our contributions as follows. Our code is available at https://github.com/ZixuanJiang/pre-rmsnorm-transformer.

- We achieve the **first-ever unification** of LayerNorm and RMSNorm in pre-normalization Transformers with proven arithmetic equivalence.

- We propose two variants: Pre-RMSNorm and Pre-CRMSNorm Transformers. The original Pre-LN Transformer and our proposed two variants are **equivalent** and can seamlessly interchange without affecting functionality.

- Our proposed architectures are $1\% - 10\%$ more **efficient** than the original Pre-LN Transformer for both training and inference. Such efficiency gains are effortlessly obtained without the need for fine-tuning or calibration.

## 2 Background

We introduce LayerNorm, RMSNorm, and their usage in Transformers. We provide an abstraction for Pre-LN Transformers.

### 2.1 LayerNorm and RMSNorm

**Layer Normalization** (LayerNorm, LN) [1] is a technique to normalize the activations of intermediate layers of neural networks. Given a vector $\boldsymbol{x} \in \mathbb{R}^d$, LayerNorm normalizes it to obtain a zero-mean unit-variance vector,

$$\text{LayerNorm}(\boldsymbol{x}) = \frac{\boldsymbol{x} - \mu(\boldsymbol{x})\mathbf{1}}{\sqrt{\|\boldsymbol{x}\|_2^2/d - \mu^2(\boldsymbol{x}) + \epsilon}}, \text{ where } \mu(\boldsymbol{x}) = \frac{\mathbf{1}^T \boldsymbol{x}}{d}, \epsilon > 0. \tag{1}$$

LayerNorm recenters and rescales the activations and gradients in the forward and backward computations [41], which enables fast and robust training of neural networks.

**Root Mean Square Normalization** (RMSNorm) [44] is another technique used for normalizing the activations. It is similar to LayerNorm in that it aims to accelerate and stabilize the training but uses a different normalization approach. Instead of normalizing the inputs based on their mean and variance, RMSNorm normalizes them based on their root mean square (RMS) value. It is defined in the following equation,

$$\text{RMSNorm}(\boldsymbol{x}) = \frac{\boldsymbol{x}}{\sqrt{\|\boldsymbol{x}\|_2^2/d + \epsilon}}, \text{ where } \epsilon > 0. \tag{2}$$

RMSNorm only rescales the input vector and the corresponding gradients, discarding the recentering process. As shown in their definitions, RMSNorm is computationally simpler and more efficient than LayerNorm. It is reported that replacing LayerNorm with RMSNorm can achieve comparable performance and save training and inference time by $7\% - 64\%$ [44].

Given a zero-mean vector $\boldsymbol{x}$, these two kinds of normalization are equivalent. Formally, if $\mu(\boldsymbol{x}) = 0$, then $\text{LayerNorm}(\boldsymbol{x}) = \text{RMSNorm}(\boldsymbol{x})$. We may optionally introduce learnable parameters and apply an element-wise affine transformation on the output of LayerNorm and RMSNorm.

We focus on the computation of LayerNorm and RMSNorm instead of their optimization and expressivity in this paper. [2]

### 2.2 Normalization in Transformers

Normalization plays a crucial role and has many variants in Transformers [39]. LayerNorm is widely used in Transformer architectures to address this issue. The position of LN within the architecture is essential for the final performance. While the initial Transformer uses Post-LN, most Transformers employ Pre-LN to achieve more stable training, even though this can result in decreased performance [40]. Pre-LN is the mainstream normalization in Transformers, especially the large models, such as ViT [13, 9], PaLM [8], and GPT-series models [30, 5].

RMSNorm is proposed as an alternative normalization technique in Transformers. Several large language models, such as Chinchilla [16] and LLaMA [38], use Pre-RMSNorm in their blocks [45]. RMSNorm can help accelerate the training and inference with similar performance in these large models. Specifically, the experiments in [26] show that RMSNorm improves the pre-training speed by $5\%$ compared with the LayerNorm baseline.

---

[2]The following discussion on expressivity is out of the scope of this paper and is only for reference. Normalization helps accelerate and stabilize the training but hurts the model's expressivity. For example, we usually add learnable parameters $\gamma, \beta$ after normalization to recover the model expressivity. LayerNorm removes the mean and variance information of the input vector, such that all the output must have zero mean and unit variance. RMSNorm only removes the scale information, so it has less information loss.

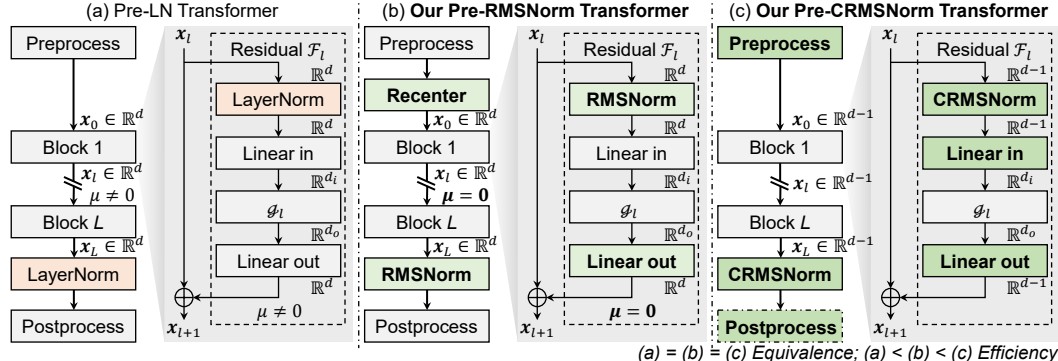

Figure 2: **Left.** The original Pre-LN Transformer architecture. **Middle and Right.** Our proposed Pre-RMSNorm and Pre-CRMSNorm Transformer architectures. These three architectures are equivalent. The differences are highlighted in bold and green blocks.

It is challenging to convert Transformers with one normalization to the other type. It is not clear which version of normalization is more suitable for Transformers.

### 2.3 Pre-LN Transformer

Figure 2.a illustrates the Pre-LN Transformer architecture, which consists of three parts, preprocessing, a stack of $L$ blocks, and postprocessing.

**Preprocessing.** We preprocess the raw inputs, which range from paragraphs in natural language [39], images [13], to state-action-reward trajectories in reinforcement learning problems[7]. We import special tokens (such as the classification token) and embeddings (such as positional embeddings), ultimately obtaining a sequence of token embeddings $x_0$.

**Transformer blocks.** The main body of Pre-LN Transformer [40] is a stack of residual blocks

$$x_{l+1} = x_l + \mathcal{F}_l(x_l), \; l = 0, 1, ..., L-1, \tag{3}$$

where $x_l$ is the input of the $l$-th block, $\mathcal{F}_l$ is a sequence of operators *LayerNorm → Linear → $g_l$ → Linear*. We name $x_l$ as the vectors on the main branch and $\mathcal{F}_l$ as the residual branch [14]. The block $\mathcal{F}_l$ is usually an attention or a multi-layer perceptron (MLP) module. If $g_l$ is an activation function, such as GELU [15], then the block $\mathcal{F}_l$ is a two-layer MLP. If $g_l$ is a (masked) multi-head scaled dot product attention, then the block is the (casual) attention module [39]. These two linear layers are usually explicitly defined. Taking the attention module as an example, the input linear projection generates the query, key, and value vectors, while the output projection is applied to the concatenated results of all heads. If they are not explicitly defined, we can add an identity mapping, a special linear layer. Most of the learnable parameters of Transformers are in these two linear layers. [3]

**LayerNorm and postprocessing.** We finally process the result $\text{LN}(x_L)$ to obtain the task-related results, such as classification probabilities. We apply the layer normalization on $x_L$ since it usually has a high variance because it is the accumulation of all the Transformer blocks.

## 3 Method

We propose Pre-RMSNorm and Pre-CRMSNorm Transformer variants, shown in Figure 2, and claim that Pre-LN, Pre-RMSNorm, and Pre-CRMSNorm Transformers are arithmetically equivalent.

$$\texttt{Pre-LN Transformer} = \texttt{Pre-RMSNorm Transformer} = \texttt{Pre-CRMSNorm Transformer}. \tag{4}$$

We will show the equivalence of these three architectures and then analyze the computational efficiency improvement by our proposed model variants. We discuss the Post-LN in Appendix B.

---

[3]The elementwise affine transformation of LayerNorm or RMSNorm is also a linear transformation. Thus, it can be fused with the input linear projection. We disable this transformation in the normalization layers to simplify the analysis.

To clarify, our proposed Pre-RMSNorm Transformer and the pre-existing Pre-RMSNorm models (e.g., LLaMA [38]) are different. Ours has a recentering and special output linear layer, as shown in Figure 2b. However, they can share the same implementation for inference. We adopt the term "Pre-RMSNorm Transformer" to represent our proposed one in the discussions below.

## 3.1 Pre-LN Transformer = Pre-RMSNorm Transformer

LayerNorm is invariant to the shifting $\text{LN}(\boldsymbol{x} + k\mathbf{1}) = \text{LN}(\boldsymbol{x}), \forall k \in \mathbb{R}$. We observe that LayerNorm is applied to the main branch vectors before they are used in either residual branches or the final postprocessing in Pre-LN Transformers. Therefore, we can replace the main branch vectors $\boldsymbol{x}_l$ with $\boldsymbol{x}_l + k_l\mathbf{1}, \forall k_l \in \mathbb{R}$ without impact on the functionality of the Pre-LN Transformer. If $k_l = -\mu(\boldsymbol{x}_l)$, we replace the main branch vectors with its recentered version $\boldsymbol{x}_l - \mu(\boldsymbol{x}_l)\mathbf{1}$. We can explicitly maintain zero-mean main branches with the same arithmetic functionality.

We propose three modifications to the original Pre-LN Transformer to obtain an equivalent Pre-RMSNorm Transformer.

1. Recenter the $\boldsymbol{x}_0$ before the first Transformer block, where $\text{Recenter}(\boldsymbol{x}) = \boldsymbol{x} - \mu(\boldsymbol{x})\mathbf{1}$.
2. For the output projection in residual branches, replace the weight $\boldsymbol{A}_o$ and bias $\boldsymbol{b}_o$ with $\hat{\boldsymbol{A}}_o = \boldsymbol{A}_o - \frac{1}{d}\mathbf{1}\mathbf{1}^T\boldsymbol{A}_o, \hat{\boldsymbol{b}}_o = \boldsymbol{b}_o - \mu(\boldsymbol{b}_o)\mathbf{1}$, where $d$ is the dimension of $\boldsymbol{x}_0$.
3. Replace LayerNorm with RMSNorm at the beginning of residual blocks and before postprocessing.

Since $\mu(\boldsymbol{x}_{l+1}) = \mu(\boldsymbol{x}_l) + \mu(\mathcal{F}_l(\boldsymbol{x}_l))$, we can keep zero-mean on the main branch *if and only if* the input of the first block $\boldsymbol{x}_0$ and the output of each residual branch $\mathcal{F}_l$ are re-centered with zero-mean. The first modification is to recenter $\boldsymbol{x}_0$, while the second modification is to recenter the output of residual branches. For the residual branch $\mathcal{F}_l$, the ending linear transformation enables us to recenter its output without extra computation, implied by Lemma 3.1. We can recenter the weight and bias of a linear layer to recenter its output.

**Lemma 3.1** *Given a linear transformation $\boldsymbol{y} = \boldsymbol{A}\boldsymbol{x} + \boldsymbol{b}, \boldsymbol{x} \in \mathbb{R}^n, \boldsymbol{A} \in \mathbb{R}^{m \times n}, \boldsymbol{b}, \boldsymbol{y} \in \mathbb{R}^m$, we can decompose the output with two parts $\boldsymbol{y} = \hat{\boldsymbol{A}}\boldsymbol{x} + \hat{\boldsymbol{b}} + \mu(\boldsymbol{y})\mathbf{1}$. The first part $\hat{\boldsymbol{A}}\boldsymbol{x} + \hat{\boldsymbol{b}} = \boldsymbol{y} - \mu(\boldsymbol{y})\mathbf{1}$, with zero mean, is another linear transformation with $\hat{\boldsymbol{A}} = \boldsymbol{A} - \frac{1}{m}\mathbf{1}\mathbf{1}^T\boldsymbol{A}, \hat{\boldsymbol{b}} = \boldsymbol{b} - \mu(\boldsymbol{b})\mathbf{1}$.*

The first two modifications ensure that we maintain zero mean on the main branch vectors. Given a zero-mean input, LayerNorm is equivalent to RMSNorm, which implies that the third modification has no impact on the functionality. With these modifications, we demonstrate that Pre-LN and Pre-RMSNorm Transformers are equivalent.

## 3.2 Pre-RMSNorm Transformer = Pre-CRMSNorm Transformer

For a zero-mean vector $\boldsymbol{x} \in \mathbb{R}^d$, we can compress it losslessly by discarding its last element. In the decompression, we recover the discarded element with $x_d = -\sum_{i=0}^{d-1} x_i$. The decompression has an extra cost, while the compression does not induce extra computation. The space-saving ratio of this compression method is $1/d$.

We define **Compressed Root Mean Square Normalization** (CRMSNorm), which takes a vector $\boldsymbol{x} \in \mathbb{R}^{d-1}$ as input. CRMSNorm first decompresses the vector $\boldsymbol{x}$ to obtain a zero-mean vector in $\mathbb{R}^d$, then applies RMSNorm on the zero-mean vector. It can generate the normalized zero-mean results in either $\mathbb{R}^{d-1}$ or $\mathbb{R}^d$. Its formal definition is in Equation 5.

$$\text{CRMSNorm}(\boldsymbol{x}) = \frac{\boldsymbol{x}}{\sqrt{(\sum_{i=1}^{d-1} x_i^2 + (\sum_{i=1}^{d-1} x_i)^2)/d + \epsilon}}, \text{ where } \boldsymbol{x} \in \mathbb{R}^{d-1}. \tag{5}$$

We simplify the Pre-RMSNorm Transformers to obtain the Pre-CRMSNorm Transformers with the following modifications.

1. Compress the zero-mean main-branch vectors from $\mathbb{R}^d$ to $\mathbb{R}^{d-1}$. Preprocessing and postprocessing handle compressed vectors in $\mathbb{R}^{d-1}$.

2. Replace RMSNorm with CRMSNorm.

3. Simplify the weight $A_i$ in the input projection layer. Let $a_d$ be the last column vector of $A_i$. $\hat{A}_i = A_i - a_d \mathbf{1}^T$ is the compressed weight matrix.

4. Simplify the weight and bias in the output projection layer. We discard the last row of the weight matrix $\hat{A}_o$ and the last element of the bias $\hat{b}_o$ since they are used to generate the redundant last element.

We compress the zero-mean activations in our proposed Pre-RMSNorm Transformers and correspondingly simplify the two linear layers in residual branches.

We can fuse preprocessing, recentering, and compression. The fused preprocessing generates compressed vectors in $\mathbb{R}^{d-1}$ to represent zero-mean vectors in $\mathbb{R}^d$. Taking language models as an example, we can recenter and compress (discard the last element) the word and position embeddings.

For the input linear projection, its input is the output of RMSNorm, whose mean is zero. We compress the output of the RMSNorm and the weight of the linear layer. Specifically, if the linear layer takes a zero-mean vector as input, then

$$\text{Linear}(\boldsymbol{x}) = \boldsymbol{A}_i \boldsymbol{x} + \boldsymbol{b}_i = (\boldsymbol{A}_i - \boldsymbol{a}_d \mathbf{1}^T)\boldsymbol{x} + \boldsymbol{b}. \tag{6}$$

$\hat{A}_i = A_i - a_d \mathbf{1}^T$ is the compressed weight matrix since its last column is a zero vector. The simplified linear layer only needs the compressed zero-mean vectors in $\mathbb{R}^{d-1}$ as input.

For the output linear projection, we only need to calculate the first $(d-1)$ elements for the vectors in $\mathbb{R}^d$. Thus, we can compress the weight $\hat{A}_o$ and bias $\hat{b}_o$. As shown in Figure 2, the shape of $\hat{A}_o$ is $(d, d_o)$, we can compress it directly to $(d-1, d_o)$ by discarding its last row. Similarly, we can compress $\hat{b}_o \in \mathbb{R}^d$ by discarding its last element. This weight compression also reflects their redundancy. As shown in Section 3.1, $\hat{b}_o$ and all column vectors of $\hat{A}_o$ are zero-mean.

Figure 2 demonstrates the difference in vector dimension. We demonstrate the equivalence between Pre-RMSNorm and Pre-CRMSNorm Transformers.

### 3.3 Pre-LN Transformer = Pre-CRMSNorm Transformer

To translate a pre-trained Pre-LN Transformer to a Pre-CRMSNorm Transformer, we can convert it into a Pre-RMSNorm model and then finish the conversion. For the model implementation, we need two steps to convert a Pre-LN Transformer to Pre-CRMSNorm Transformer.

1. Reduce the hidden dimension from $d$ to $d-1$. [4]

2. Replace LayerNorm with CRMSNorm.

Namely, Pre-LN Transformers with the main branch vectors in $\mathbb{R}^d$ are equivalent to Pre-CRMSNorm Transformers in $\mathbb{R}^{d-1}$, which further echoes the redundancy in the Pre-LN Transformers.

### 3.4 Training and Inference Efficiency

We show how to make conversions between the three variants. The conversions only consist of one-time parameter adjustments without expensive fine-tuning or calibration, similar to operator fusion [27]. We qualitatively analyze the efficiency improvement of our proposed Pre-(C)RMSNorm Transformers compared with equivalent Pre-LN models.

#### 3.4.1 Pre-RMSNorm Transformer

We discuss the impact of three modifications on training and inference, as shown in Table 1.

**The linear layer with zero-mean output.** The modified linear layer will not induce extra computation for inference since we can replace the weight and bias in advance. During inference, we can treat it as a standard linear layer with equivalently transformed parameters.

---

[4]We reduce the hidden dimension in the preprocessing, main branch, two linear layers in residual branches, and postprocessing. We keep the $d_i, d_o$ in residual branches. For instance, we maintain the MLP dimension and head dimension (dimension of query, key, and value vectors).

| | Recenter | Zero-Mean Linear | RMSNorm |
|---|---|---|---|
| **Training** | a little increase | a little increase | decrease |
| **Inference** | same | same | |

Table 1: The computation workload of our Pre-RMSNorm model compared with the original Pre-LN Transformer.

We have to pay the extra cost for training for the parameter change. The parameter optimizer manages $\boldsymbol{A}_o, \boldsymbol{b}_o$, but we use their recentered version $\hat{\boldsymbol{A}}_o, \hat{\boldsymbol{b}}_o$ during training. The induced cost is small for several reasons. (1) The size of parameters $\boldsymbol{A}_o, \boldsymbol{b}_o$ is relatively small since they do not depend on the batch size and sequence length. For reference, the computation cost of LayerNorm in the original Pre-LN Transformer is proportional to the batch size and the sequence length. (2) Obtaining $\hat{\boldsymbol{A}}_o, \hat{\boldsymbol{b}}_o$ can be done ahead of time since it does not depend on the input. We may leverage the idle time of accelerators to compute them. (3) In data-parallel distributed training, the parameter server [19] manages the parameters. Each worker will receive $\hat{\boldsymbol{A}}_o, \hat{\boldsymbol{b}}_o$ from the server and then pass the gradients of loss to $\hat{\boldsymbol{A}}_o, \hat{\boldsymbol{b}}_o$ to the server. Only the server needs to maintain and update the original $\boldsymbol{A}_o, \boldsymbol{b}_o$. In short, it is much easier to process the model parameters than the intermediate activations.

**Recentering.** It is possible to fuse the recentering with the preprocessing, which usually handles the sum of several kinds of embeddings. For example, the input embeddings of the BERT model [11] are the accumulation of the token embeddings, the segmentation embeddings, and the position embeddings. Since $\text{Recenter}(\boldsymbol{x}+\boldsymbol{y}) = \text{Recenter}(\boldsymbol{x})+\text{Recenter}(\boldsymbol{y})$, recentering the input is equivalent to recentering each embedding before the addition. We can recenter the accumulated embeddings or each embedding separately before the accumulation. Suppose an embedding is from a linear layer. In that case, we can modify the linear layer such that it generates the zero-mean output, similar to how we edit the output linear projection.

For inference, we can recenter the related embeddings or linear layers in advance such that no extra computation is induced. For training, recentering induces extra cost since it is on the fly.

**Replacing LayerNorm with RMSNorm.** Section 2.1 introduces that RMSNorm can achieve speedup compared with LayerNorm, as demonstrated by the previous models. This replacement can help us accelerate the training and inference of the Pre-LN Transformer.

### 3.4.2 Pre-CRMSNorm Transformer

CRMSNorm, an extension of RMSNorm, saves execution time as it is more computationally efficient than LayerNorm. Additionally, CRMSNorm further reduces the hidden dimension from $d$ to $d-1$, which can reduce the model size, computation, communication, and memory consumption by $1/d$ in theory. However, most accelerators can not efficiently handle the vectors in $\mathbb{R}^{d-1}$ when $d$ is a large even number, especially a power of two (e.g., 1024, 4096). This limitation arises because these accelerators are typically optimized for arithmetic with even dimensions. In some cases, handling $\mathbb{R}^{d-1}$ vectors may take much more time than $\mathbb{R}^d$ vectors. Thus, we need to examine if the accelerators can support $\mathbb{R}^{d-1}$ vectors efficiently. If not, we have the following alternatives. For inference, we may either (1) add zero embeddings, or (2) decompress the vectors and translate the model into the Pre-RMSNorm variant. For training, we suggest keeping the hidden dimension $d$, which is equivalent to a Pre-LN model with the hidden dimension $d+1$. In this way, the CRMSNorm can help us increase the model representability and computation efficiency at the same time.

### 3.5 Training and Inference Equivalence

Taking Pre-LN and Pre-RMSNorm Transformers as examples, we further explain the arithmetic equivalence.

**Inference equivalence.** Let $f$ be a Pre-LN Transformer with parameter $\theta$, and $g$ be a Pre-RMSNorm Transformer with parameter $\phi$. We demonstrate that for any input $\boldsymbol{x} \in \mathbb{R}^d$, we can always have $f(\boldsymbol{x}, \theta) = g(\boldsymbol{x}, \phi = h(\theta))$ and we show how we conduct the conversion $\phi = h(\theta)$. Thus, we claim that $f$ and $g$ are equivalent in terms of arithmetic functionality. Namely, when calculating $f(\boldsymbol{x}, \theta)$ during inference, we can always compute the equivalent counterpart $g(\boldsymbol{x}, \phi = h(\theta))$. Our

proposed method is a re-parameterization technique [12], which builds equivalent models with different parameters.

**Training equivalence.** Despite $f(\boldsymbol{x}, \theta) = g(\boldsymbol{x}, \phi = h(\theta))$, there may be a large difference in the convergence speed and stability when training $f$ and $g$ with SGD and its variants. We avoid this issue by applying the gradient updates on $\theta$ instead of $\phi$. Specifically, during the training process, we maintain a Pre-LN Transformer model $f$ and its parameter $\theta$. In the forward and backward computation, we convert the model and its parameter into $g$ and $\phi$ to save training time. We do not update $\phi$ directly. Instead, we calculate the gradients $\nabla\theta$ and call the SGD optimizer $\theta = \theta - \eta\nabla\theta$, where $\eta$ is the learning rate. From the perspective of optimization, the training is still on the Pre-LN Transformer model $f$ and its parameter $\theta$. Consequently, the training performance and stability are preserved and equivalent to the original Pre-LN Transformers.

# 4 Experiments

We claim that our major contributions are the unification and equivalence of the three Transformer variants. We do not report the task-related performance, such as the classification accuracy and perplexity, since the training and inference equivalence are guaranteed. We discuss how we verify the task-related performance in Appendix D.4.

The efficiency is a free lunch accompanying the equivalence. Now that the efficiency of RMSNorm over LayerNorm has been shown in the previous work [44, 26], we focus on analyzing the efficiency of each component of our method.

We conduct experiments on ViT [13, 36] and GPT-3 [5] since they represent two mainstream architectures of Transformers, encoder-only and casual decoder. Other Transformer variants can be treated as an extension of these two architectures, such as encoder-decoder [39], and non-causal decoder [43]. Also, ViT and GPT cover the areas of computer vision and natural language, where Transformers have been popular and achieved great success.

We abstain from utilizing pre-trained weights. We replicate the ViT and GPT-3 architectures as described in their respective papers, employing dummy data to assess the training and inference speed. We use PyTorch 2.0 [28] to build the training and inference pipeline. We run iterations at least 100 times and report the 25th, 50th (median), and 75th percentile since these quartiles are more robust than the mean. We use the automatic mixed precision [24] for both training and inference.

## 4.1 Experiments on ViT

The ViT takes images with 3 channels and a resolution of $224 \times 224$ and generates a classification over 1000 classes, which is the standard setting for ImageNet training and inference.

In the training recipe of ViT [13], dropout [33] is added at the end of each residual branch, which breaks the zero-mean property of the residual output. Hence, in Pre-RMSNorm Transformers, we need to recenter the output vectors explicitly at the end of the residual branch. [5] The Pre-CRMSNorm Transformers are compatible with dropout since it uses vectors in $\mathbb{R}^{d-1}$ to represent the compressed zero-mean vectors in $\mathbb{R}^d$. On the contrary, DeiT [36] disables the dropout and can guarantee the zero-mean output, in spite that the stochastic depth [17] and LayerScale [37] are applied. We follow the settings in DeiT [36] in this paper.

**Inference.** Figure 3 illustrates the inference time with different batch sizes and model sizes on a single A100 GPU. Taking the Pre-LN Transformer as the baseline, our proposed Pre-RMSNorm Transformer can reduce the inference time by $1\% - 9\%$. This reduction takes effect for ViTs with different models and various mini-batch sizes. The left subfigure demonstrates the inference latency, which is the inference time when the batch size is 1.

The proportion of LayerNorm in the total computation is $12\% - 18\%$ in these experiments. As discussed in Sections 2.1 and 3.4.1, replacing LayerNorm with RMSNorm can help us accelerate the inference. RMSNorm can reduce the inference time of LayerNorm by $20\% - 60\%$, thus helping us achieves a stale faster inference for Pre-RMSNorm models.

---

[5] Dropout only impacts training and has no impact on inference. Thus, the Pre-RMSNorm variant can also be used for inference.

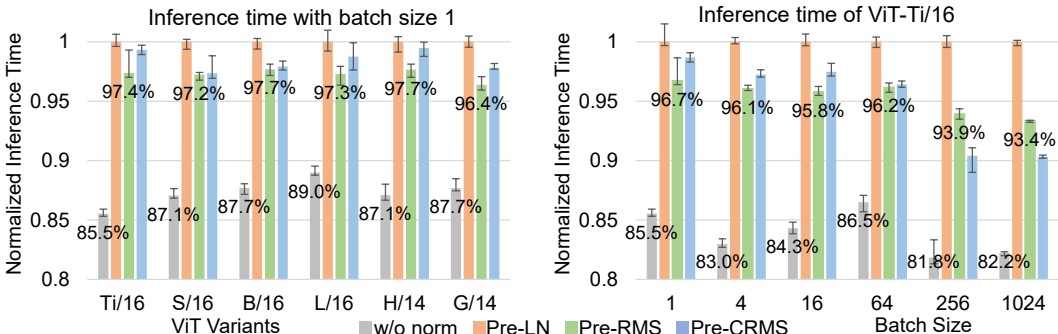

Figure 3: Normalized inference time on ViT with different model sizes and batch sizes.

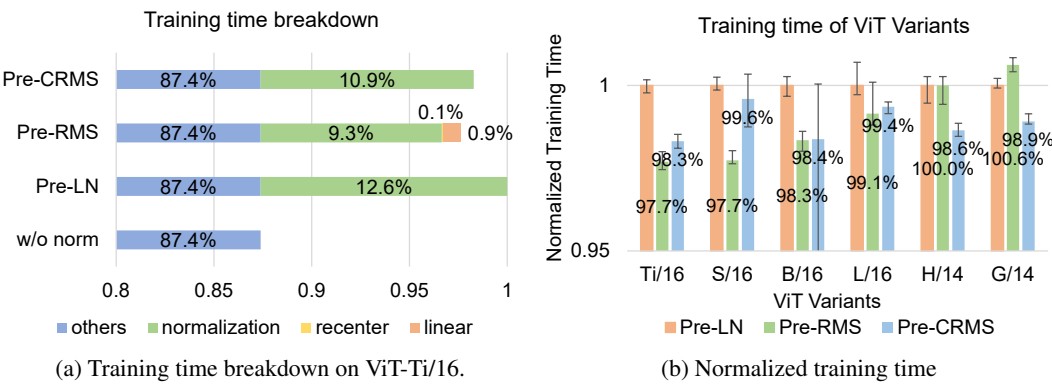

(a) Training time breakdown on ViT-Ti/16.

(b) Normalized training time

Figure 4: Training time comparison and breakdown on ViT variants

For Pre-CRMSNorm, the GPU cannot efficiently handle vectors in $\mathbb{R}^{d-1}$ in some cases, where we add zero padding to obtain vectors in $\mathbb{R}^d$. With zero padding, Pre-CRMSNorm models are less efficient than Pre-RMSNorm ones due to the extra decompression. However, for the cases where $\mathbb{R}^{d-1}$ vectors can be accelerated efficiently, we can achieve an even $10\%$ time reduction.

We also conduct experiments (1) with other precisions, (2) on CPUs, (3) with JAX [4] and observe similar performance. For these ViT variants, we have achieved an average of $3.0\%$ inference time reduction. Please see Appendix D for more details.

**Training.** We train ViTs on a single workstation with 4 A100s with data parallel training [20], following the DeiT training recipe. Each training iteration consists of forward and backward computation on all the workers, gradient all-reduce, parameters update with an optimizer, and broadcasting the new parameters. Figure 4a visualizes the breakdown of the related components. In the Pre-LN Transformer, the layer normalization accounts for $12.6\%$ of total training time. For the Pre-RMSNorm variant, we have to modify the output projection and recenter the input, which induces the $0.89\%$ and $0.09\%$ extra computation time. Then LayerNorm is reduced to RMSNorm, whose computation cost is $9.27\%$. Overall, the Pre-RMSNorm reduces the training time by $2.36\%$.

We train the Pre-CRMSNorm Transformers with $d$ as the hidden dimension since we do not obtain a speedup from the compressed dimension since the GPUs cannot handle $\mathbb{R}^{d-1}$ vectors efficiently. The CRMSNorm is more computationally expensive than the RMSNorm given the same input but takes less time than LayerNorm. The Pre-CRMSNorm Transformer does not need the recentering and special linear layers to generate zero-mean results at the end of residual branches. Above all, training the Pre-CRMSNorm variant is $1.74\%$ faster than the Pre-LN model.

Figure 4b illustrates the training time of different ViTs. We have achieved a speedup of $1\% - 2.5\%$, which is smaller than the inference speedup. Considering only the forward and backward computation, the speedup is similar between training and inference. Nevertheless, the training needs extra time on gradient all-reduce, optimizer update, and parameters broadcast, which shrinks the percentage of

normalizations in the whole computation. For reference, the percentage is $10\% - 15\%$ for training these ViTs.

## 4.2 Experiments on GPT

We measure the inference time on the GPT-3 [5] variants with batch size 1 and sequence length 512. The results are shown in Figure 5. In small GPT-3 models, layer normalization takes considerable time. Hence, replacing the Layer-Norm with (C)RMSNorm can reduce the inference time by up to $10\%$. However, as the model grows, the attention and MLP take charge of the main part of the computation [18]. The normalization takes $< 1\%$ of the inference time for GPT-3 XL and larger models, which is the upper bound of the speedup with our methods.

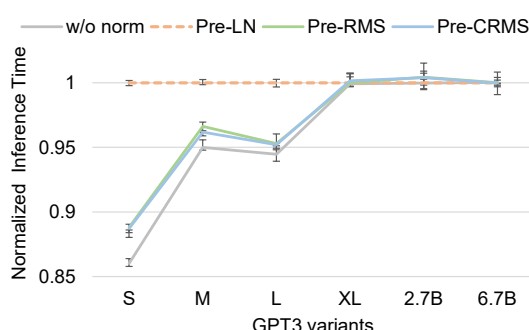

Figure 5: GPT-3 inference performance

Applying quantization may mitigate this issue to some extent. By applying int8 matrix multiplication [10], the percentage of the layer normalization increases to $10\%$ for GPT-3 XL and 2.7B. Our Pre-RMSNorm can reduce the inference time by $4\%$. Training performance is similar to the ViT one. We have achieved $1.5\%$ and $1.8\%$ time reduction for GPT-3 Small and Medium, respectively.

## 5 Conclusion

In this paper, we propose two equivalent and efficient variants for the widely used Pre-LN Transformers. We point out the inherent redundancy in the Pre-LN Transformer. By maintaining zero-mean on the main branch vectors and thus simplifying LayerNorm, we obtain the Pre-RMSNorm architecture. We further apply a lossless compression on the zero-mean vectors to obtain the Pre-CRMSNorm model. For the first time, We unify these normalization variants within the Transformer model.

We enable the more efficient utilization of Pre-LN Transformers, allowing for seamless transitions between normalization techniques with minimal overhead. We can replace a Pre-LN Transformer with an equivalent Pre-(C)RMSNorm Transformer with better training and inference efficiency, which is a free lunch. We strictly push the performance-efficiency Pareto frontier of foundational Pre-LN Transformers. As a result, pre-trained Pre-LN Transformers (such as ViT and GPT) can be deployed more efficiently, and new equivalent or superior models can be trained with higher throughput.

**Extensions.** We believe that our proposed CRMSNorm technique has the potential for application in other neural architectures. Further exploration of its usage in different contexts would be beneficial. Additionally, while our focus has been on pre-normalization Transformers, it would be valuable to investigate the application of our methods to other related architectures, especially the foundation models. Finally, integrating our proposed method into machine learning compilers could directly enable the generation of equivalent and simplified computation graphs.

**Limitations.** Detailed implementations and specific optimizations will play a crucial role in achieving practical performance gains. It is necessary to focus on developing highly optimized implementations of (C)RMSNorm to bridge the gap between theoretical and practical efficiency improvements. By addressing these challenges, we can fully leverage the potential benefits of (C)RMSNorm and enable more efficient utilization of Pre-LN Transformers.

## Acknowledgments and Disclosure of Funding

We acknowledge NVIDIA for donating its A100 GPU workstations and the support from TILOS, an NSF-funded National Artificial Intelligence Research Institute.

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

# Appendices

## A  Proof for Lemma 1.

Given a linear transformation $\boldsymbol{y} = \boldsymbol{A}\boldsymbol{x} + \boldsymbol{b}, \boldsymbol{x} \in \mathbb{R}^n, \boldsymbol{A} \in \mathbb{R}^{m \times n}, \boldsymbol{b}, \boldsymbol{y} \in \mathbb{R}^m$, we have

$$\boldsymbol{y} = \boldsymbol{A}\boldsymbol{x} + \boldsymbol{b} \tag{7}$$

$$= (\boldsymbol{A} - k\mathbf{1}\mathbf{1}^T\boldsymbol{A})\boldsymbol{x} + k\mathbf{1}\mathbf{1}^T\boldsymbol{A}\boldsymbol{x} + (\boldsymbol{b} - \mu(\boldsymbol{b})\mathbf{1}) + \mu(\boldsymbol{b})\mathbf{1} \tag{8}$$

$$= (\boldsymbol{A} - k\mathbf{1}\mathbf{1}^T\boldsymbol{A})\boldsymbol{x} + (\boldsymbol{b} - \mu(\boldsymbol{b})\mathbf{1}) + k(\mathbf{1}^T\boldsymbol{A}\boldsymbol{x})\mathbf{1} + \mu(\boldsymbol{b})\mathbf{1} \tag{9}$$

$$= (\boldsymbol{A} - k\mathbf{1}\mathbf{1}^T\boldsymbol{A})\boldsymbol{x} + (\boldsymbol{b} - \mu(\boldsymbol{b})\mathbf{1}) + (k\mathbf{1}^T\boldsymbol{A}\boldsymbol{x} + \mu(\boldsymbol{b}))\mathbf{1} \tag{10}$$

$$= \hat{\boldsymbol{A}}\boldsymbol{x} + \hat{\boldsymbol{b}} + f(\boldsymbol{x}, k)\mathbf{1} \tag{11}$$

where $\hat{\boldsymbol{A}} = \boldsymbol{A} - k\mathbf{1}\mathbf{1}^T\boldsymbol{A}, \hat{\boldsymbol{b}} = \boldsymbol{b} - \mu(\boldsymbol{b})\mathbf{1}, f(\boldsymbol{x}, k) = k\mathbf{1}^T\boldsymbol{A}\boldsymbol{x} + \mu(\boldsymbol{b})$.

If $k = 1/m$, then we obtain

$$\mu(\hat{\boldsymbol{A}}\boldsymbol{x}) = \frac{1}{m}\mathbf{1}^T(\boldsymbol{A} - \frac{1}{m}\mathbf{1}\mathbf{1}^T\boldsymbol{A})\boldsymbol{x} = \frac{1}{m}(\mathbf{1}^T\boldsymbol{A} - \mathbf{1}^T\boldsymbol{A})\boldsymbol{x} = 0 \tag{12}$$

$$\mu(\boldsymbol{y}) = \mu(\hat{\boldsymbol{A}}\boldsymbol{x}) + \mu(\hat{\boldsymbol{b}}) + \mu(f(\boldsymbol{x}, k = 1/m)\mathbf{1}) \tag{13}$$

$$= 0 + 0 + f(\boldsymbol{x}, k = 1/m) \tag{14}$$

$$= \frac{1}{m}\mathbf{1}^T\boldsymbol{A}\boldsymbol{x} + \mu(\boldsymbol{b}) \tag{15}$$

The $\hat{\boldsymbol{A}}$ is the recentered matrix of $\boldsymbol{A}$, and all its column vectors have zero-mean. We decompose the output into two parts.

- The first part $\hat{\boldsymbol{A}}\boldsymbol{x} + \hat{\boldsymbol{b}} = \boldsymbol{y} - \mu(\boldsymbol{y})\mathbf{1}$, with zero mean, is another linear transformation with $\hat{\boldsymbol{A}} = \boldsymbol{A} - \frac{1}{m}\mathbf{1}\mathbf{1}^T\boldsymbol{A}, \hat{\boldsymbol{b}} = \boldsymbol{b} - \mu(\boldsymbol{b})\mathbf{1}$.
- The second part corresponds to the mean information $\mu(\boldsymbol{y})\mathbf{1} = (\frac{1}{m}\mathbf{1}^T\boldsymbol{A}\boldsymbol{x} + \mu(\boldsymbol{b}))\mathbf{1}$.

## B  Post-LN Transformers

Different from the Pre-LN Transformers, the Post-LN Transformers have the following blocks.

$$\boldsymbol{x}_{l+1} = \text{LN}(\boldsymbol{x}_l + \mathcal{F}_l(\boldsymbol{x}_l)), \ l = 0, 1, ..., L - 1, \tag{16}$$

Layer normalization is on the main branch instead of the beginning of residual branches. We can keep a zero-mean branch on the main branch without impacting the functionality.

$$\boldsymbol{x}_{l+1} = \text{LN}(\boldsymbol{x}_l + \mathcal{F}_l(\boldsymbol{x}_l)) \tag{17}$$

$$= \text{LN}((\boldsymbol{x}_l - \mu(\boldsymbol{x}_l)\mathbf{1}) + (\mathcal{F}_l(\boldsymbol{x}_l) - \mu(\mathcal{F}_l(\boldsymbol{x}_l))\mathbf{1})) \tag{18}$$

$$= \text{LN}(\hat{\boldsymbol{x}}_l + \hat{\mathcal{F}}_l(\boldsymbol{x}_l)) \tag{19}$$

$$= \text{RMSNorm}(\hat{\boldsymbol{x}}_l + \hat{\mathcal{F}}_l(\boldsymbol{x}_l)) \tag{20}$$

For the residual branch $\mathcal{F}_l$, we can apply the same method in the Pre-LN Transformer. We can modify the output linear projection to obtain $\hat{\mathcal{F}}_l$, which generates the zero-mean part of the original result.

The recentering operation $\hat{\boldsymbol{x}}_l = \boldsymbol{x}_l - \mu(\boldsymbol{x}_l)\mathbf{1}$ requires extra computation. If elementwise affine transformation is disabled in LayerNorm, $\boldsymbol{x}_l$ is the output of a normalization such that $\mu(\boldsymbol{x}_l) = 0$ and $\hat{\boldsymbol{x}}_l = \boldsymbol{x}_l$. If the transformation is enabled, $\boldsymbol{x}_l$ is not guaranteed zero-mean such that explicit recentering is necessary.

## C  Revertible Conversions

The conversions of Pre-LN → Pre-RMSNorm and Pre-RMSNorm → Pre-CRMSNorm are listed in Sections 3.1 and 3.2. These steps are fully reversible. We write the inverse steps explicitly below.

**Coverting Pre-RMSNorm into Pre-LN Transformers.**

1. Remove the recentering.

2. $\boldsymbol{A}_o = \hat{\boldsymbol{A}}_o + \boldsymbol{1}\boldsymbol{c}^T$, where $\boldsymbol{c}$ can be a random vector, $\boldsymbol{b}_o = \hat{\boldsymbol{b}}_o + d\boldsymbol{1}$ where d can be a random scalar.

3. Replace RMSNorm with LayerNorm.

**Coverting Pre-CRMSNorm into Pre-RMSNorm Transformers.**

1. Decompression vectors in $\mathbb{R}^{d-1}$ into zero-mean vectors in $\mathbb{R}^d$. The compression is lossless, so the decompression is its invertible operation.

2. Replace CRMSNorm with RMSNorm.

3. $\boldsymbol{A}_i = \hat{\boldsymbol{A}}_i + \boldsymbol{a}_d\boldsymbol{1}^T$, where $\boldsymbol{a}_d$ is a random vector.

4. Recover the last row of $\hat{\boldsymbol{A}}_o$ and the last element of $\hat{\boldsymbol{b}}_o$ such that each column of $\hat{\boldsymbol{A}}_o$ and $\hat{\boldsymbol{b}}_o$ is zero-mean.

# D  Experiments

## D.1  Implementation of Normalization

We have provided our implementation with JAX and PyTorch in the supplementary material. The reported results are based on the following implementations.

For JAX, we use the APIs of LayerNorm and RMSNorm in the flax library. For PyTorch, we use the implementations of LayerNorm and RMSNorm from NVIDIA's apex extension [6]. For CRMSNorm, we use our own customized implementations. We also provide our customized implementation of LayerNorm and RMSNorm.

We notice that there are lots of APIs for the standard LayerNorm and RMSNorm. For example, PyTorch has provided the official LayerNorm API but lacks RMSNorm implementation. These different implementations are mixed. We do not find one implementation dominant over others for all the cases. For instance, `torch.nn.LayerNorm` is usually faster than `apex.normalization.FusedLayerNorm` when the input vectors are small in inference, while it is slower than the apex when the input vectors are large. PyTorch's official implementation is also slower than the apex for training.

## D.2  Extended Experiments in ViT

| Name | Dimension | Depth | Heads | MLP Dimension |
|---|---|---|---|---|
| Tiny-16 | 192 | 12 | 3 | $192 \times 4$ |
| Small-16 | 384 | 12 | 6 | $384 \times 4$ |
| Base-16 | 768 | 12 | 12 | $768 \times 4$ |
| Large-16 | 1024 | 24 | 16 | $1024 \times 4$ |
| Huge-14 | 1280 | 32 | 16 | $1280 \times 4$ |
| Giant-14 | 1664 | 48 | 16 | 8192 |

Table 2: ViTs with different sizes. The number in the model name is the patch size.

| | no norm | Pre-LN | Pre-RMS | Pre-CRMS |
|---|---|---|---|---|
| **PyTorch, single A100, amp** | 0.8567 | 1.000 | 0.9699 | 0.9783 |
| amp $\rightarrow$ float32 | 0.9353 | 1.000 | 0.9850 | 0.9951 |
| single A100 $\rightarrow$ 16-thread CPU | 0.8697 | 1.000 | 0.9012 | 0.8857 |
| PyTorch $\rightarrow$ JAX | 0.9610 | 1.000 | 0.9873 | 1.0005 |

Table 3: Normalized inference time of ViT.

Table 2 lists the architecture parameters of Vision Transformer. We first measure the **inference time**. We sweep these 6 ViTs with 6 batch sizes (1, 4, 16, 64, 256, 1024) and collect the medians of these

---

[6]https://github.com/NVIDIA/apex

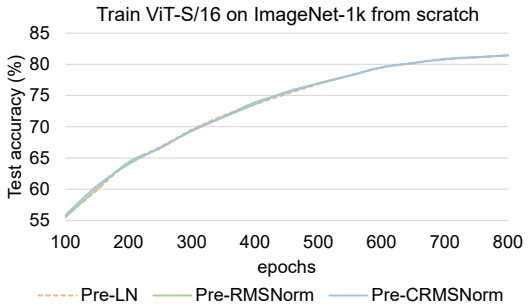

Figure 6: The test accuracy of training a ViT-S/16 on ImageNet-1k from scratch.

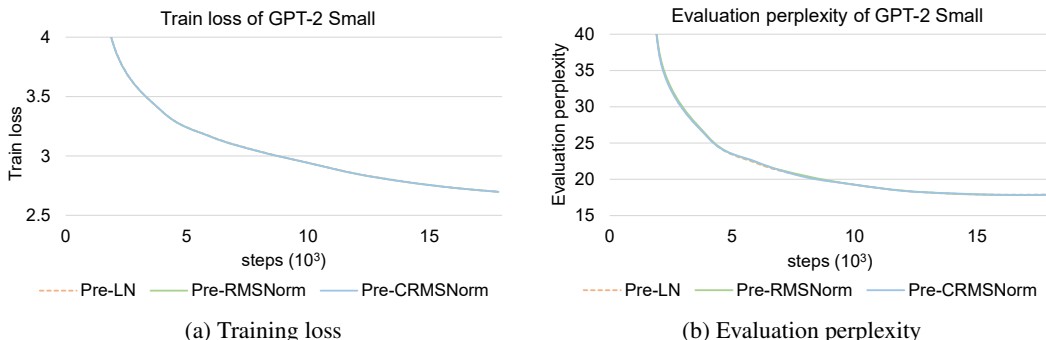

(a) Training loss

(b) Evaluation perplexity

Figure 7: We train GPT-2 Small on the wikitext-103-raw-v1 dataset from scratch.

36 data points. We report the average of these 36 experiments in Table 3. We conduct inference on a single A100 with automatic mixed precision (amp) in PyTorch. We further change the precision (disabling the amp), computation platforms (16 threads in AMD EPYC 7742 CPUs), and machine learning frameworks (JAX).

### D.3 Numerical Issue

The theoretical arithmetic equivalence cannot be fully translated into equality in practical numerical computation if we use floating numbers. An intuitive example is that $\mu(\boldsymbol{x}+\boldsymbol{y}) = \mu(\boldsymbol{x})+\mu(\boldsymbol{y})$ always holds for any vectors $\boldsymbol{x}, \boldsymbol{y}$. However, if these two vectors are represented as (low precision) floating numbers, this equality is not guaranteed in real-world numerical computation. It is possible that these small discrepancies may be accumulated and enlarged in the large models, further degrading the numerical stability. In our proposed method, we cannot ensure the exact zero-mean in the main branch numerically.

The numerical issue is a common problem in machine learning. A typical example is operator reordering and layer fusion. PyTorch provides a related API officially, named `torch.ao.quantization.fuse_modules`. We can fuse the convolution layer and its following batch normalization layer to simplify the computation. These two layers are separate in training and can be fused to accelerate the inference. The fusion does not break the arithmetic equivalence but changes the numerical results. In spite of the numerical difference, the fusion usually has a neutral impact on task-related performance, such as classification accuracy, even in large models. Fine-tuning or calibration may be helpful in case there is severe performance degradation.

Our proposed methods encounter a similar issue as layer fusion since we modify partial parameters. In our experiments, we can convert the pre-trained Pre-LN ViT-H/14 into Pre-(C)RMS variants without any accuracy change on the ImageNet validation dataset. We observe that replacing PyTorch's official LayerNorm implementation with the apex may have a larger impact on the model performance.

## D.4 Verification of Training Equivalence

For validation and reference, we train ViT-S/16 as Pre-LN, Pre-RMSNorm, and Pre-CRMSNorm models on ImageNet-1k following the DeiT-3 setting and achieve similar accuracy. The test accuracy over 800 epochs is shown in Figure 6. Remarkably, these Transformer variants yield similar training performance. We also train three variants of GPT-2 Small (12 layers, 768 hidden dimensions) on the wikitext-103-raw-v1 dataset [23] from scratch. The training and evaluation losses are demonstrated in Figure 7. These performance curves exhibit remarkable similarity, providing empirical evidence for their training equivalence.

