# OpenReview forum: "Pre-RMSNorm and Pre-CRMSNorm Transformers: Equivalent and Efficient Pre-LN Transformers"
_NeurIPS.cc/2023/Conference — NeurIPS 2023 spotlight_

### Official Review · Reviewer_NSno · 2023-06-16

**Soundness:** 1 poor
**Presentation:** 3 good
**Contribution:** 3 good
**Rating:** 7
**Confidence:** 3

**Summary:**

The paper analyses the two common types of normalization layers in Transformers: LayerNorm and RMSNorm (root-mean-square-norm). LayerNorm scales all vectors to be of the same norm, while changing the vectors' "directions". RMSNorm, in contrast, keeps the same direction, but just rescales the vectors to the same constant norm.
The authors "unify" both approaches, by showing how to force LayerNorm to act like RMSNorm. Then, since RMSNorm is faster to compute, the authors argue that this conversion (LayerNorm -> RMSNorm) can get the expressivity benefits of LayerNorm with the speed of RMSNorm.
Finally, the authors propose C-RMSNorm, which saves 1 dimension. For example, if the model hidden size is 1024, C-RMSNorm allows reducing it to 1023 while calculating the norm.
Overall, the authors argue that these changes can reduce the training and inference time of LayerNorm transformers by 10% (although without discussing whether there's a reduction in performance/accuracy).

**Strengths:**

## Strengths
+ The authors provide a thorough explanation of LayerNorm vs. RMSNorm.
+ The authors show the relation between LayerNorm vs. RMSNorm, which I haven't seen in any other paper.
+ The proposed approaches can sometimes reduce inference and training time, which is an important problem that solving can save millions and save carbon emmission.

**Weaknesses:**

## Weaknesses
- I'm not sure about the soundness of some of the claims. The authors argue that LayerNorm is **equivalent** to RMSNorm and even write explicitly `PreLN Trasnformer = Pre-RMSNorm Transformer` (Eq 4). However, they only show one direction of this equality: the authors show how to take a Pre-LayerNorm transformer, and by imposing specific values to weights and enforcing constraints it can be equivalent to RMSNorm. That is, by constraining LayerNorm you can achieve RMSNorm. To my understanding, this means that Pre-LayerNorm is **at least** as expressive as Pre-RMSNorm. But what about the other direction? Maybe Pre-LayerNorm is more expressive if we do not impose these constraints? Can we take a Pre-RMSNorm and make it a Pre-LayerNorm?
- Similarly, I am not sure that the authors prove both directions when claiming in Section 3.2 that `Pre-RMSNorm = Pre-C-RMSNorm`.
- Novelty - The authors claim to "propose Pre-RMSNorm". Is this proposal novel? aren't models such as LLaMA already using Pre-RMSNorm?
- Another issue with soundness: the authors write multiple times that these conversions are "free efficiency improvement" and "free lunch". However, in Line 228 they write: "We have to pay the extra cost for training for the parameter change". So, what exactly do the authors mean by "free lunch" if there is an extra training cost, and the accuracy is not guaranteed to be the same?
- Many evaluation details are missing.
    - On which datasets do the authors perform the experiments?
    - How can the authors perform training and inference with **GPT-3**? As far as I know, GPT-3 is not open-sourced. Further, the authors mention they used "**GPT3 XL and 2.7B**". What exactly do they mean here? What is "**GPT3 XL**"? and what is "**GPT3 2.7B**", Is there such a model? The sizes of GPT-3 that I know about are much bigger than 2.7B. Where exactly this model was taken from and what exactly do the authors mean?

**Questions:**

## Questions
1. Section 3.1 says that "LayerNorm is applied to the main branch vectors before they are used in either residual branches or the final postprocessing. Therefore, we can replace the main branch vectors $x_l$ with $x_l + k_l\mathbb{1}$" - I am not sure I agree. I agree that since LayerNorm is invariant to the shifting $LN(x) = LN(x_l + k_l\mathbb{1})$, the replacement will not change the output of the **residual branch**. However, if we replace the main branch vectors as suggested, the summation of the main branch and the residual branch that happens at the end of each transformer layer - is no longer identical to its original version.
2. In most of the cases in Figures 3+4, C-RMSNorm is slower than RMSNorm. So what is the benefit of C-RMSNorm?
3. Evaluation - the evaluation compares only the speed of training/inference of different normalization layers, without discussing the result of the training. Maybe Pre-RMSNorm is faster than Pre-LayerNorm, but needs to be trained for 10% **more steps** to reach the same training loss? Maybe even after training longer, it does not achieve the same test accuracy? I will not be surprised if that's the case: since the authors showed that RMSNorm is a special case of LayerNorm, I would expect LayerNorm to be more expressive and possibly reach better results, or reach these results within fewer steps.
4. On which datasets were the experiments conducted? What was the downstream accuracy of each model?
5. What exactly are the GPT-3 models mentioned in the evaluation?
6. Are the claims and findings in this paper related to the findings in the following papers regarding LayerNorm?

    * ["On the Expressivity Role of LayerNorm in Transformers' Attention", Brody et al., EMNLP'2023](https://lessw.medium.com/what-layernorm-really-does-for-attention-in-transformers-4901ea6d890e)
    * ["Understanding and Improving Layer Normalization", Xu et al., NeurIPS 2019](https://arxiv.org/pdf/1911.07013.pdf)
    * ["On Layer Normalization in the Transformer Architecture", Xiong et al., ICML 2020](https://arxiv.org/pdf/2002.04745.pdf)
I'm not asking this question to make the authors just cite these papers, I am actually trying to understand whether this paper is consistent with, is orthogonal to, or contradicts these papers, since these are some of the most significant papers that were written about the analysis of LayerNorm that I know of, and none of them is discussed.

## Summary
Overall, the paper provides a thorough analysis and comparison of LayerNorm and RMSNorm, and provides insights that lead to 10% reduced inference and training time. If these improvements are valid and real, a 10% reduction is significant and important.
However, there are many questions regarding the validity of the evaluation and the soundness of the claims. Therefore, I'm currently voting for a borderline reject, until the authors clarify all the issues above. I will raise my score if all issues are explained and resolved.

---

> ### Author Rebuttal · Authors · 2023-08-07
>
> Thanks a lot for your insightful comments. Due to the word limit, we have to redirect you to our response to other reviewers. We sincerely appreciate your understanding.
>
> ### **Question 1**
>
> The original Transformer block is
> * $x_{l+1}=x_l+F_l(x_l)$
> * $x_{l+2}=x_l+F_l(x_l)+F_{l+1}(x_l+F_l(x_l))$
>
> $F_l(x_l) = F_l(x_l + k_l \mathbb{1})$ since $F_l$ begins with a LayerNorm. The proposed block with shifting is
> * $y_{l+1} = y_l+k_l \mathbb{1}+F_l(y_l+k_l\mathbb{1})=y_l+F_l(y_l)+k_l\mathbb{1}$
> * $y_{l+2} = y_l+F_l(y_l) + F_{l+1}(y_l+F_l(y_l))+(k_l+k_{l+1})\mathbb{1}$
>
> The main branch vectors are different $y_l=x_l+\sum_{i=0}^l k_l\mathbb{1}$. However, the difference has no impact on arithmetic functionality. $y_l$ is used in residual branches and final postprocessing. These two blocks begin with a LayerNorm, so they are invariant to this difference.
>
> We also give an example to show the arithmetic equivalence in our submitted code.
>
> ### **Question 2**
>
> Please check our response to **Reviewer AcCQ on Vectors in $\mathbb{R}^{d-1}$ on accelerators**.
>
> ### **Question 3**
>
> Please check our response to **Reviewer j4qQ on Weaknesses 3 and 1**. In short, we maintain and update Pre-LN model parameters to guarantee training equivalence.
>
> ### **Question 4, Weakness 5.1**
> Since we guarantee training equivalence, we do not focus on datasets and accuracy.
>
> For validation and reference, we train ViT-S/16 as a Pre-RMSNorm and a Pre-CRMSNorm model on ImageNet-1k following the DeiT-3 setting and achieve similar accuracy (81.5%, 81.4% for our models, and 81.4% for baseline).
>
> ### **Question 5, Weakness 5.2**
> GPT-2 is publicly available at https://github.com/openai/gpt-2. In Section 2.1 of the GPT-3 paper [1], the authors describe that
> > We use the same model and architecture as GPT-2, including the modified initialization, pre-normalization, and reversible tokenization described therein, ……, we train 8 different sizes of model, ……
>
> The hyperparameters of these 8 models (Small, Medium, Large, XL, 2.7B, 6.7B, 13B, 175B) are defined in Table 2.1 of the GPT-3 paper. We follow the naming and description in their paper to build GPT-3 models.
>
> [1] Brown, Tom, et al. "Language models are few-shot learners." NeurIPS 2020.
>
> ### **Question 6**
>
> We focus on the computation of LayerNorm and RMSNorm without touching their optimization and expressivity. We discuss how we calculate them rather than their role in optimization. Hence, our work is orthogonal and complementary to your mentioned work, which focuses on expressivity [2], the effect on optimization [3], and the location [4] of LN in Transformers. We have also discussed two of them.
> * Lines 94-95. LN takes effect in both forward and backward computation [3].
> * Lines 109-114. Pre-LN Transformers are easier to optimize than Post-LN variants [4], which is the major reason that Pre-LN is much more popular than Post-LN. Hence, we focus on Pre-LN architectures and only discuss Post-LN in Appendix B.
>
> [2] On the Expressivity Role of LayerNorm in Transformers' Attention. EMNLP 2023.
>
> [3] Understanding and Improving Layer Normalization. NeurIPS 2019.
>
> [4] On Layer Normalization in the Transformer Architecture. ICML 2020.
>
> ### **Weaknesses 1, 2, and 3**
>
> We first define two modifications. M1: add a Recenter after Preprocess. M2: adjust parameters of the output linear layer in residual branches to generate zero-mean output. We define four models.
> * LN: The general Pre-LN model in Figure 2a
> * LN-M: Pre-LN with two modifications M1 & M2
> * RMS: The existing Pre-RMSNorm model, e.g., LLaMA
> * RMS-M: Our proposed Pre-RMSNorm model with M1 & M2 in Figure 2b
>
> We prove that `LN = LN-M = RMS-M`, where `=` means arithmetic equivalence. `LN = LN-M` because the mean information is redundant on the main branch. The modifications do not hurt the expressivity of Pre-LN models. `LN-M = RMS-M` because `LN(x) = RMSNorm(x)` if x is zero-mean.
>
> `RMS` and `RMS-M` are not equivalent. Model definitions of them for training are different since M1 and M2 need to be handled explicitly. Conversely, they can share the same implementation for inference since M1 can be fused with preprocessing, and M2 can be done before inference (Section 3.4.1).
>
> **Model expressivity.** Although the expressivity is beyond the scope of our paper, we provide a new perspective on that.
> * If the two modifications (M1 & M2) harm the model expressivity, the expressivity ranking is `LN = LN-M = RMS-M < RMS`.
> * If M1 & M2 are helpful, the ranking is `LN = LN-M = RMS-M > RMS`.
>
> **Model conversions in the inverse direction.**
> Pre-LN -> Pre-RMS needs 3 steps (Lines 156-160), and Pre-RMS -> Pre-CRMS needs 4 steps (Lines 185-192). These steps are fully reversible. We write the inverse steps explicitly below.
>
> Pre-RMS -> Pre-LN
> 1. Remove the recentering
> 2. $A_o=\hat{A_o} + \mathbb{1}c^T$, where c can be a random vector, $b_o=\hat{b_o}+d\mathbb{1}$ where d can be a random scalar
> 3. RMSNorm -> LayerNorm
>
> Pre-CRMS -> Pre-RMS
> 1. Decompression vectors in $\mathbb{R}^{d-1}$ into zero-mean vectors in $\mathbb{R}^{d}$. The compression is lossless, so the decompression is its invertible operation.
> 2. CRMSNorm -> RMSNorm
> 3. $A_i=\hat{A_i}+a_d\mathbb{1}^T$, where $a_d$ is a random vector
> 4. Recover the last row of $\hat{A_o}$ and the last element of $\hat{b_o}$ such that each column of $\hat{A_o}$ and $\hat{b_o}$ is zero-mean.
>
> `{LN, LN-M, RMS-M, CRMS}` can be converted in any direction. `RMS` cannot be converted to and from them.
>
> **Normalization.** For zero-mean vector, the equation `LayerNorm(x) = RMSNorm(x)` holds. For any vector x, we only have `LayerNorm(x) = RMSNorm(Recenter(x))`.
>
> ### **Weakness 4**
> Please check our response to **Reviewer AcCQ on Slight increase in training workload**. In short, the extra training cost is a local overhead, while the efficiency improvement refers to a global reduction.
>
> Thanks again for reviewing our paper. We are anticipating further discussions with you.

---

> > ### Comment · Reviewer_NSno · 2023-08-13
> > **Response to authors**
> >
> > Thank you for your response.
> >
> > >Q3. our contributions: identifying the inherent redundancy in Pre-LN Transformers
> >
> > I am not sure I agree. Pre-LN Transformers change the "direction" of the vector, while Pre-RMS don't.
> > I thus argue that Pre-RMS is inherently less expressive than Pre-LN.
> >
> > In some special cases, when the input is "centered" in a certain way, they are equivalent, but I was not convinced that this holds for the general case.
> >
> > > Q4. Since we guarantee training equivalence, we do not focus on datasets and accuracy.
> >
> > Even if there is equivalence in theory, I would like to be sure that there is no empirical degradation in practice.
> >
> > Why can't you show training and test curves?
> >
> > Are all training loss and test loss curves completely identical?
> >
> >
> > > Q5. GPT-2 is publicly available at https://github.com/openai/gpt-2 ...
> >
> > So, do you mean that this is just the 2.7B architecture, which can be referred to as either GPT-2 or GPT-3?
> > Did you use the pretrained weights, or just the bare architecture which you trained from scratch?
> >
> > ## Weaknesses
> > I am not sure I am following, since the explanations in your response do not use the same terms as in the paper.
> >
> > I still don't understand how can you claim equivalence, and write explicitly that `PreLN Transformer = Pre-RMSNorm` , if one of the directions applies only when the input is recentered?
> >
> > Moreover, can you explicitly refer to my question in the original review:
> > >Novelty - The authors claim to "propose Pre-RMSNorm". Is this proposal novel? aren't models such as LLaMA already using Pre-RMSNorm?

---

> > > ### Author Response · Authors · 2023-08-13
> > > **Thank you for your response**
> > >
> > > Thank you so much for your insightful response. We are delighted that most of your comments have been addressed. The answers to the remaining questions are listed below.
> > >
> > > ### **Weakness**
> > > We use {Pre-LN, Pre-RMS} in our paper and {LN, LN-M, RMS, RMS-M} in our response to demonstrate their relationship more clearly. We are sorry for the confusion and will refine our paper accordingly. The definitions in our rebuttal are
> > > >* LN: The general Pre-LN model in Figure 2a
> > > >* LN-M: Pre-LN with two modifications M1 & M2
> > > >* RMS: The existing Pre-RMSNorm model, e.g., LLaMA
> > > >* RMS-M: Our proposed Pre-RMSNorm model with M1 & M2 in Figure 2b
> > >
> > > The Pre-LN, Pre-RMS in the paper corresponds to `LN` and `RMS-M`. The modifications represented by the suffix `M` enforce a zero-mean main branch. Our equivalence is `LN = LN-M = RMS-M`.
> > > * `LN = LN-M` because the mean information is redundant on the main branch. The modifications do not impact the arithmetic functionality.
> > > * `LN-M = RMS-M` because `LN(x) = RMSNorm(x)` if x is zero-mean.
> > >
> > > Please note that `RMS` is not equivalent to {`LN`, `LN-M`, `RMS-M`}. Due to the relationship `LN = LN-M = RMS-M != RMS`, {`LN`, `LN-M`, `RMS-M`} can be converted in any direction. `RMS` cannot be converted to and from them. For instance, we can convert `RMS-M` into `LN`, but we cannot alter an `RMS` into `LN`.
> > >
> > > **Novelty of Pre-RMSNorm models.**
> > > Our proposed models are `RMS-M`, while the existing models (such as LLaMA) are `RMS`. The novelty is that we propose a new Transformer variant with RMSNorm that are equivalent to Pre-LN Transformer. `RMS-M` and `RMS` can share the same inference implementation.
> > >
> > > ### **Q3, inherent redundancy**
> > >
> > > **Computation.** The main branch is used in blocks starting from LayerNorm. Thus, its mean information is redundant and can be removed without impacting the computation result. The term **inherent redundancy** refers to the unnecessary computation of the mean information.
> > >
> > > Our paper focuses on how we compute the normalization instead of their role in model expressivity. The following discussion is out of the scope of our paper and is only used for reference.
> > >
> > > **Expressivity.** From our understanding, normalization helps accelerate and stabilize the training but hurts the model expressivity. For example, we usually add learnable parameters $\gamma, \beta$ after normalization to recover the model expressivity. LayerNorm removes the mean and variance information of the input vector, such that all the output must have zero mean and unit variance. RMSNorm only removes the scale information, so it has less information loss.
> > >
> > > ### **Q4**
> > > The loss curves are not completely identical for two reasons.
> > > 1. Randomness. Training models with the same settings cannot achieve the same results if the determinism is enabled and perfectly supported.
> > > 2. Numerical issue. It is a common problem in machine learning. A typical example is operator reordering and layer fusion. We discuss this in Appendix C.3.
> > >
> > > We find that OpenReview is not allowed to edit the global response. We are trying to figure out how to attach the figure of training curves in the general response.
> > >
> > > ### **Q5**
> > > We follow the naming and model architectures in the GPT-3 paper. We do not have the pre-trained weights and dataset. We build the architecture following the GPT-3 paper and use dummy data to evaluate the speed for training and inference.
> > >
> > > Once again, we appreciate your insightful and timely response. We are looking forward to further discussions with you.

---

> > > > ### Comment · Reviewer_NSno · 2023-08-13
> > > > **Thank you for the clarification**
> > > >
> > > > Thank you for the clarifications.
> > > >
> > > > >We will attach the figure of training curves in the general response in 12 hours.
> > > >
> > > > I would be convinced that there is no practical degradation only if you include **the test curves as well**.
> > > > For language modeling, this can be the perplexity on a separate validation set.
> > > >
> > > > >We follow the naming and model architectures in the GPT-3 paper. We do not have the pre-trained weights and dataset. We build the architecture following the GPT-3 paper and use dummy data to evaluate the speed for training and inference.
> > > >
> > > > I thus recommend not to call it "GPT-3". When readers read "GPT-`X`" the assumption is that you refer to the actual trained weights, the actual checkpoint that is called GPT-3 (which is not publicly available).
> > > > This is implied from the current phrasing in the text:
> > > > `We conduct experiments on ViT [13, 34] and GPT3`.
> > > > Instead, I suggest writing something like: `"We conduct experiments with a GPT-3-based architecture"`.
> > > >
> > > > Thanks again for the clarifications.
> > > > I will increase my score if the training and test curves of the GPT model will be convincing.

---

> > > > > ### Author Response · Authors · 2023-08-13
> > > > > **Thank you for your fast response**
> > > > >
> > > > > Thank you for the fast feedback.
> > > > >
> > > > > We find that authors are not allowed to edit the global response at this stage. We are trying to figure out how to attach the figure of training and test curves. We have already contacted the committee for help.
> > > > >
> > > > > We will follow your great suggestions on the GPT-3 to polish our paper.
> > > > >
> > > > > We sincerely appreciate all your comments.

---

> > > > > > ### Comment · Reviewer_NSno · 2023-08-13
> > > > > > **Another option**
> > > > > >
> > > > > > An alternative to visual curves could be the following:
> > > > > >
> > > > > > 1. Train the base model and the modified model on the training set of a standard language modeling dataset, such as [Wikitext](https://huggingface.co/datasets/wikitext) or any other non-synthetic dataset, for a few days.
> > > > > > 2. Report in a table (instead of a figure) the training loss/perplexity and the validation loss/perplexity, after every `N` steps.
> > > > > >
> > > > > > Thank you

---

> > > > > > > ### Author Response · Authors · 2023-08-14
> > > > > > > **The training results of ViT are provided. We are working on training a language model.**
> > > > > > >
> > > > > > > Thanks for your understanding. In our response, we mention that
> > > > > > > > For validation and reference, we train ViT-S/16 as a Pre-RMSNorm and a Pre-CRMSNorm model on ImageNet-1k following the DeiT-3 setting and achieve similar accuracy.
> > > > > > >
> > > > > > > The test accuracy after every 100 epochs is listed in the table. This is the result of a single run.
> > > > > > >
> > > > > > > | test acc (%) at epoch   | 100   | 200   | 300   | 400   | 500   | 600   | 700   | 800   |
> > > > > > > |--------------|-------|-------|-------|-------|-------|-------|-------|-------|
> > > > > > > | Pre-LN       | 55.62 | 64.25 | 69.54 | 73.59 | 76.84 | 79.48 | 80.80 | 81.39 |
> > > > > > > | Pre-RMSNorm  | 55.50 | 64.34 | 69.47 | 73.93 | 76.99 | 79.49 | 80.89 | 81.46 |
> > > > > > > | Pre-CRMSNorm | 55.84 | 63.99 | 69.31 | 73.60 | 76.90 | 79.52 | 80.79 | 81.41 |
> > > > > > >
> > > > > > > We also train a small customized ViT on CIFAR-100 10 times for each model variant. The statistics (avg $\pm$ std) of training loss and test accuracy are listed in the table.
> > > > > > >
> > > > > > > | train loss at epochs       | 50           | 100          | 150          | 200          | 250          | 300          |
> > > > > > > |--------------|--------------|--------------|--------------|--------------|--------------|--------------|
> > > > > > > | Pre-LN       | 3.149 $\pm$ 0.065 | 2.747 $\pm$ 0.068 | 2.435 $\pm$ 0.103 | 2.260 $\pm$ 0.095 | 2.072 $\pm$ 0.091 | 2.041 $\pm$ 0.008 |
> > > > > > > | Pre-RMSNorm  | 3.117 $\pm$ 0.059 | 2.725 $\pm$ 0.070 | 2.450 $\pm$ 0.099 | 2.209 $\pm$ 0.100 | 2.054 $\pm$ 0.095 | 2.034 $\pm$ 0.008 |
> > > > > > > | Pre-CRMSNorm | 3.156 $\pm$ 0.067 | 2.774 $\pm$ 0.064 | 2.489 $\pm$ 0.105 | 2.272 $\pm$ 0.095 | 2.070 $\pm$ 0.091 | 2.046 $\pm$ 0.008 |
> > > > > > >
> > > > > > > | test acc (%) at epoch | 50          | 100         | 150         | 200         | 250         | 300         |
> > > > > > > |--------------|-------------|-------------|-------------|-------------|-------------|-------------|
> > > > > > > | Pre-LN       | 56.98 $\pm$ 0.89 | 65.88 $\pm$ 0.72 | 69.52 $\pm$ 0.63 | 72.07 $\pm$ 0.36 | 74.06 $\pm$ 0.32 | 74.81 $\pm$ 0.33 |
> > > > > > > | Pre-RMSNorm  | 56.74 $\pm$ 0.95 | 65.94 $\pm$ 0.70 | 69.60 $\pm$ 0.62 | 72.15 $\pm$ 0.35 | 74.17 $\pm$ 0.34 | 74.95 $\pm$ 0.32 |
> > > > > > > | Pre-CRMSNorm | 57.11 $\pm$ 0.88 | 65.92 $\pm$ 0.75 | 69.48 $\pm$ 0.63 | 72.01 $\pm$ 0.35 | 74.15 $\pm$ 0.33 | 74.84 $\pm$ 0.31 |
> > > > > > >
> > > > > > > The results show that training performance is similar across the three model variants. The difference comes from randomness and the numerical issue.
> > > > > > >
> > > > > > > We are working on training a small language model on real datasets. We will update the results in several days. Thank you again for your engaging and enlightening discussions.

---

> > > > > > > > ### Author Response · Authors · 2023-08-17
> > > > > > > > **The training results on GPT-2 small on the real dataset are available**
> > > > > > > >
> > > > > > > > Thank you for your continued help. Following the examples provided by huggingface/transformers (https://github.com/huggingface/transformers/tree/main/examples/pytorch/language-modeling), we train three variants of GPT-2 Small (12 layers, 768 hidden dimensions) on the **wikitext-103-raw-v1** dataset from scratch. We report the results of a single run.
> > > > > > > >
> > > > > > > > The train loss every 2000 steps is listed in the table.
> > > > > > > >
> > > > > > > > | train loss at step        | 2k    | 4k    | 6k    | 8k    | 10k   | 12k   | 14k   | 16k   | 17,850 |
> > > > > > > > |--------------|-------|-------|-------|-------|-------|-------|-------|-------|--------|
> > > > > > > > | Pre-LN       | 3.920 | 3.369 | 3.162 | 3.038 | 2.941 | 2.849 | 2.783 | 2.730 | 2.698  |
> > > > > > > > | Pre-RMSNorm  | 3.914 | 3.367 | 3.161 | 3.038 | 2.942 | 2.853 | 2.782 | 2.727 | 2.698  |
> > > > > > > > | Pre-CRMSNorm | 3.916 | 3.370 | 3.160 | 3.036 | 2.941 | 2.848 | 2.781 | 2.731 | 2.697  |
> > > > > > > >
> > > > > > > > The evaluation loss every 2000 steps is listed below. The final evaluation perplexity is 17.8979, 17.8535, 17.8749 for Pre-LN, Pre-RMSNorm, Pre-CRMSNorm models. The training performance is similar across the three variants.
> > > > > > > >
> > > > > > > > | eval loss at step        | 2k    | 4k    | 6k    | 8k    | 10k   | 12k   | 14k   | 16k   | 17,850 |
> > > > > > > > |--------------|-------|-------|-------|-------|-------|-------|-------|-------|--------|
> > > > > > > > | Pre-LN       | 3.630 | 3.252 | 3.101 | 3.014 | 2.958 | 2.914 | 2.892 | 2.884 | 2.885  |
> > > > > > > > | Pre-RMSNorm  | 3.639 | 3.256 | 3.106 | 3.019 | 2.957 | 2.914 | 2.891 | 2.881 | 2.882  |
> > > > > > > > | Pre-CRMSNorm | 3.627 | 3.250 | 3.109 | 3.010 | 2.959 | 2.912 | 2.894 | 2.883 | 2.883  |
> > > > > > > >
> > > > > > > > Moreover, the conference committee told us that "the rebuttal PDF should have been updated during the rebuttal stage and cannot be uploaded now." Hence, we extend our genuine gratitude for your suggestion to present the outcomes through tables.
> > > > > > > >
> > > > > > > > Thank you again for reviewing our paper and response. Please reach out to us if you have any questions.

---

> > > > > > > > > ### Comment · Reviewer_NSno · 2023-08-17
> > > > > > > > > **Thank you**
> > > > > > > > >
> > > > > > > > > Thank you for the additional experiments and for your efforts!
> > > > > > > > >
> > > > > > > > > I have increased my score to 7. Please include all additional details and experiments as part of the final paper.

---

> > > > > > > > > > ### Author Response · Authors · 2023-08-17
> > > > > > > > > > **Thank you for your insights**
> > > > > > > > > >
> > > > > > > > > > We really appreciate your comments and discussions, which are helpful for us to refine the paper!

---

### Official Review · Reviewer_j4qQ · 2023-07-06

**Soundness:** 3 good
**Presentation:** 3 good
**Contribution:** 3 good
**Rating:** 5
**Confidence:** 5

**Summary:**

This study explores the relationship between Pre-RMSNorm and Pre-LN Transformers, demonstrating that these two variants can be theoretically reparameterized into one another. Additionally, the authors introduce a novel Transformer variant called Pre-CRMSNorm, which reduces one hidden dimension while maintaining the same representation power as Pre-LN and Pre-RMSNorm. Experimental evaluations on ViT and GPT models reveal that the proposed approach significantly improves computation efficiency during both model training and inference stages.

**Strengths:**

The findings are interesting and reasonable. The proposed approach is novel and demonstrates a new direction to improve model efficiency via parameterization. The authors conduct experiments on both image data (i.e., ViT) and natural language (i.e., GPT).

**Weaknesses:**

1. The study would benefit from additional evaluations of the proposed method beyond computation efficiency. While the comparisons conducted on computation efficiency are useful, it is crucial to include discussions on the effectiveness of the resulting model. The absence of such results makes it challenging to assess the proposed method and its claims, particularly regarding training speedup. For example, it remains unclear whether the proposed reparameterization yields comparable training performance and stability when compared to Pre-LN Transformers.

2. The efficiency gain diminishes as the model size scales larger, which affects the empirical contribution of the proposed method. Additionally, without sufficient discussions on model performance, it is unclear whether the proposed method is compatible with low-precision data for model inference and training, which plays an important role in making deep learning efficient. Incorporating experiments with fp16 and bf16, as well as exploring the use of quantized models, would significantly enhance the paper and provide valuable insights.

3. The main finding of the study fails to provide a very deep insight beyond the reparameterization connection. Also, I feel the equivalent claim is a bit ambiguous, and would recommend the author phrase the claim as reparameterization. This is crucial because different reparameterizations for the same network can lead to substantial variations in training stability, training performance, and even inference performance. Emphasizing the reparameterization aspect would address this ambiguity.

**Questions:**

What would be the computation efficiency difference between the LayerNorm and RMSNorm from NVIDIA’s apex package and the customized implementation?

**Limitations:**

The authors mentioned the potential compatibility issue with low-precision data types but did not provide adequate discussions on this issue.

---

> ### Author Rebuttal · Authors · 2023-08-05
>
> Thank you sincerely for your insightful feedback. We appreciate the opportunity to address your questions in this response.
>
> ### **Weakness 3, reparameterization and equivalence**
> We would like to initially clarify these two terms. Let $f$ be a Pre-LN Transformer with parameter $\theta$, and $g$ be a Pre-RMSNorm Transformer with parameter $\phi$. We demonstrate that for any input $x \in \mathbb{R}^d$, we can always have $f(x,\theta) = g(x,\phi=h(\theta))$ and we show how we conduct the conversion $\phi=h(\theta)$. Thus, we claim that $f$ and $g$ are equivalent in terms of arithmetic functionality. Namely, when calculating $f(x,\theta)$, we can always compute the equivalent counterpart $g(x,\phi=h(\theta))$.
>
> We find that reparameterization in machine learning usually refers to a trick to enable the differentiability of discrete variables [1]. To avoid confusion and misunderstanding, we do not use that term.
>
> Beyond the equivalence and reparameterization, we would like to highlight one of our contributions: identifying the inherent redundancy in Pre-LN Transformers, which we believe could offer valuable insights to the community.
>
> ### **Weakness 1, impact on training performance and stability**
> Your concern is that although $f(x,\theta) = g(x,\phi=h(\theta))$, the convergence speed and stability may have a large difference with SGD and its variants (e.g., Adam). We avoid this issue by applying the gradient updates on $\theta$ instead of $\phi$.
>
> Specifically, during the training process, we maintain a Pre-LN Transformer model $f$ and its parameter $\theta$. In the forward and backward computation, we convert the model and its parameter into $g$ and $\phi$ to save training time. We do not update $\phi$ directly. Instead, we calculate the gradients $\nabla \theta$ and call the optimizer $\theta = \theta - \eta \nabla \theta$. From the perspective of optimization, the training is still on the Pre-LN Transformer model $f$ and its parameter $\theta$. Consequently, the training performance and stability are preserved and equivalent to the original Pre-LN Transformers. Therefore, our paper did not extensively discuss training performance, as the training results remain the same.
>
> In the supplementary material, we provide both training and inference implementations for your reference.
>
> ### **Weakness 2 and Limitation 1, low-precision**
> We appreciate your attention to our use of automatic mixed precision (AMP) for both training and inference (Line 276). We agree that low-precision calculation plays a pivotal role in our method's performance.
>
> Our method can benefit from low precision. With high precision in Transformers, matrix multiplications in the self-attention and MLP dominate the computation, while the normalization only takes a small portion. With a lower precision, matrix multiplications can be significantly accelerated, implying that normalization will take a larger portion. The larger percentage normalizations take, the larger reduction we can achieve.
>
> To illustrate this further, we have presented two examples in the paper.
> 1. Lines 338 - 342. By applying int8 matrix multiplication, the percentage of the layer normalization increases to 10% for GPT3 XL and 2.7B. Our Pre-RMSNorm can reduce the inference time by 4%. As a reference, the percentage and the reduction are < 1% in AMP.
> 2. Appendix C.2. We provide results of disabling AMP (i.e., using float32 precision). The normalization takes less percentage, leading to a smaller speedup with Pre-RMSNorm.
>
> ### **Question on the computation efficiency difference**
> We measure the inference time of the normalization layer on an input tensor with shape [1024, 1024]. The first 1024 is the batch size, and the second one is the dimension where normalization applies. RMSNorm is faster than LayerNorm since it does not need recentering.
>
> |  time (us) | LayerNorm | RMSNorm |
> |:----------:|:---------:|:-------:|
> |    apex    |   55.44   |  49.39  |
> | customized |   58.25   |  54.58  |
>
> Please refer to Appendix C.1 regarding implementations of normalization. Our submitted code also involves all kinds of normalization implementations in both PyTorch and JAX.
>
> Thank you sincerely for reviewing our paper and response. We are looking forward to further discussions with you.
>
> [1] Jang, E., Gu, S., & Poole, B. (2016). Categorical reparameterization with gumbel-softmax. ICLR 2017

---

> > ### Comment · Reviewer_j4qQ · 2023-08-14
> > **Rebuttal Response**
> >
> > Thanks for the clarification on weakness 1. I have updated my score.
> >
> > As to the re-parameterization, the main reason for suggesting this term is that I found the proposed method behaves in a similar way to the RepVGG line of work.
> >
> > RepVGG: Making VGG-style ConvNets Great Again

---

> > > ### Author Response · Authors · 2023-08-15
> > > **Thank you for your comment**
> > >
> > > Thank you sincerely for reviewing our paper and response.
> > >
> > > Your comments on the reparameterization are pretty constructive. Our work and RepVGG are very similar, which build equivalent models with different parameters. We will definitely add related discussions and clarify the terms in our refined paper.

---

### Official Review · Reviewer_d6Jr · 2023-07-08

**Soundness:** 4 excellent
**Presentation:** 4 excellent
**Contribution:** 4 excellent
**Rating:** 7
**Confidence:** 4

**Summary:**

This paper proposes modifications to the popular transformer architecture's normalization mechanism in order to improve efficiency without sacrificing performance. It starts out with the baseline architecture Pre-LN (LayerNorm), and derives two new architectures: Pre-RMSNorm and Pre-CRMSNorm, which are inspired by the use of RMSNorm instead of LayerNorm mechanism in certain transformers in the literature to improve efficiency without noticeably affecting performance.

The paper formally proves the arithmetic equivalence of all three architectures. It also evaluates the two proposed architectures over the baseline by conducting experiments on ViT and GPT3.  It reports modest improvements in training time (around 2%) and slightly bigger improvements in inference time (1-9%) for a variety of different hyperparameter settings.


**Strengths:**

The paper sheds light on normalization mechanisms in transformer architectures by relating two of the most popular such mechanisms: LayerNorm and RMSNorm.

By starting out with the empirical observation in the literature that RMSNorm is more efficient than LayerNorm, it derives two new architectures based on RMSNorm and proves them to be arithmetically equivalent to the LayerNorm architecture.

The empirical evaluation is quite thorough, using both ViT and GPT settings, and demonstrates modest improvements in both training and inference times across a wide range of hyperparameter settings.

Finally, the presentation is crystal clear: the paper is extremely well written even for an informed outsider to follow.


**Weaknesses:**

The single major weakness is that the savings in training and inference time is not too significant (around 1-3% in training time and about the same or slightly more in inference time, depending on the setting).

The authors acknowledge this by noting that normalization is not the elephant in the room, and that attention and MLP dominate the computation especially as the model size grows.  They even show an empirical upper bound on the speedup (which is the time taken by the normalization mechanism as a fraction of the overall time -- around 12-18%).

A minor nit: The last sentence of the abstract "Experiments demonstrate that we can reduce the training and inference time of Pre-LN Transformers by up to 10%" is a bit misleading.  I would appreciate it if you paraphrase this sentence, by 1) separating out the training and inference time numbers, and 2) providing both lower and upper bounds instead of just the 10% upper bound.

**Questions:**

None

**Limitations:**

Yes, the authors have adequately addressed the limitations.

---

> ### Author Rebuttal · Authors · 2023-08-05
>
> Thank you so much for your insightful reviews. We are truly grateful for your acknowledgment of our paper's excellence in terms of soundness, presentation, and contribution.
>
> We totally agree that our relative efficiency improvement is modest. As you correctly pointed out, our quantitative analysis shows that normalization takes a small portion of the computation. Moreover, we elaborate on three arguments to justify the significance of this improvement in Section 1 (Lines 65 - 78).
> > 1. The improvement comes from equivalence instead of tradeoff. We can strictly push the performance-efficiency Pareto frontier.
> 2. The modest relative saving can be translated into significant absolute improvement, especially when applied to widely-used Transformers.
> 3. Our method is orthogonal and complementary to most work improving efficiency. Our method can effectively benefit from other methods. For example, we have larger efficiency improvement with low precision since low precision increases the percentage of normalization (Section 4.2 and Appendix C.2).
>
> We genuinely value your suggestions and will diligently polish the paper, particularly enhancing the abstract to provide a clearer summary.
>
> Once again, we express our gratitude for your thoughtful review of our paper and response. We are looking forward to further discussions with you.

---

### Official Review · Reviewer_AcCQ · 2023-07-12

**Soundness:** 3 good
**Presentation:** 3 good
**Contribution:** 3 good
**Rating:** 7
**Confidence:** 3

**Summary:**

The paper proposes two novel modifications to the Pre-Layer Normalization (Pre-LN) Transformers, introducing the Pre-Root Mean Square Normalization (Pre-RMSNorm) and Pre-Compressed Root Mean Square Normalization (Pre-CRMSNorm) Transformers. The authors aim to improve computational efficiency by simplifying LayerNorm to RMSNorm and introducing a lossless compression technique for zero-mean vectors. They claim these changes maintain the arithmetic functionality of the original models, reducing training and inference time by up to 10%. The paper includes extensive mathematical proofs and technical diagrams to support their claims. Experiments using Vision Transformer (ViT) and GPT models demonstrate the improvements in speed.

**Strengths:**

The paper presents novel ideas in Transformer optimization, leading to improved computational efficiency. The use of compression in the Pre-CRMSNorm Transformer is particularly inventive. The detailed mathematical proofs and the experimental results provide a strong basis for the claims made, enhancing the paper's quality and clarity. Also, the potential for the method's applicability in various domains enhances the paper's significance.

**Weaknesses:**

Although the paper presents comprehensive mathematical proofs, the practical effectiveness of the proposed model is not extensively demonstrated. More empirical results across different domains and data types would further substantiate the authors' claims. The authors acknowledge that the proposed modifications might slightly increase the training workload, which could be a limitation for some applications. Lastly, the handling of vectors in $\mathbb{R}^{d-1}$ by accelerators is assumed, which might not hold true in all scenarios.

**Questions:**

1. Could the authors provide more empirical evidence of the model's performance on real-world tasks, including various data types and domains?
2. How do the proposed models handle scenarios where the input and output of certain blocks are not zero-mean?
3. Are there any specific scenarios where the proposed models might underperform compared to the original Pre-LN Transformer?

**Limitations:**

The paper acknowledges that the input and output of certain blocks are assumed to be zero-mean, which might not be the case in all scenarios. There might also be potential issues in the decompression stage of the CRMSNorm method that are not discussed. Further empirical validation is necessary to fully understand the impact of these modifications on various use cases and data types. Also, the proposed models might not be as efficient when the handling of vectors in $\mathbb{R}^{d-1}$ by accelerators is not optimal.

---

> ### Author Rebuttal · Authors · 2023-08-06
>
> Thank you so much for your insightful comments. We appreciate the opportunity to address your comments in this response.
>
> ### **Question 1, more empirical results on various domains and data types**
>
> **Domains.**
> We present experimental results on ViT and GPT3, covering the field of computer vision and natural language. These two models also represent two mainstream architectures of Transformers, encoder-only and casual decoder. Other Transformers can be treated as an extension of these two architectures, e.g., encoder-decoder and non-causal decoder.
>
> We also conduct our experiments on Decision Transformers [1] in reinforcement learning. It adapts the GPT2 model. On the OpenAI Gym benchmark, the model is small with only 3 layers, 1 attention head, and an embedding dimension of 128. Consequently, our method significantly reduces offline training time (2-6%) and testing time (5-13%).
>
> **Data Types.**
> We provide three levels of precisions in our paper.
> 1. By default, we use automatic mixed precision (AMP) for training and inference (Line 276).
> 2. Lines 338 - 342. By applying int8 matrix multiplication, the percentage of LayerNorm increases to 10% for GPT3 XL and 2.7B. Our Pre-RMSNorm can reduce the inference time by 4%. As a reference, the rate and the reduction are <1% in AMP.
> 3. Appendix C.2. We provide results of disabling AMP (i.e., using float32 precision). The normalization takes less percentage, leading to a smaller speedup with Pre-RMSNorm.
>
> We believe that our paper encompasses representative model architectures, diverse domains, and various data types. We are always open to adding more experiments if you can suggest specific settings.
>
> [1] Chen, L., Lu, K., et al. Decision transformer: Reinforcement learning via sequence modeling. NeurIPS 2021.
>
> ### **Question 2, zero-mean assumption**
> We discuss the general Pre-LN Transformer architecture without zero-mean assumption. In Figure 2a, we mark $\mu \neq 0$ in the main branch and the residual block.
>
> Our analysis of the Pre-LN Transformer uncovers that the mean information on the main branch is inherently redundant. We (1) add a Recenter after Preprocess, and (2) enforce the output linear layer to generate zero-mean output in the Pre-LN model. These two modifications can guarantee a zero-mean main branch and have no impact on arithmetic functionality. To achieve our Pre-RMSNorm Transformer, we further replace LayerNorm with RMSNorm since they are equivalent on zero-mean input. We mark $\mu = 0$ in Figure 2b.
>
> In summary, zero-mean is not an assumption in Pre-LN Transformers. It is an intentional architectural modification to achieve our desired Pre-RMSNorm models without impacting arithmetic functionality. We provide an example to show this conversion in our submitted code.
>
> ### **Question 3, when proposed models are worse than Pre-LN Transformers**
> We discuss the limitations at the end of the paper. Less-optimized implementations of (C)RMSNorm may eclipse our proposed methods. Highly optimized implementations of (C)RMSNorm are critical and will further improve our models’ efficiency.
>
> ### **Vectors in $\mathbb{R}^{d-1}$ on accelerators**
> We discuss this issue in Pre-CRMSNorm Transformer in Section 3.4.2 and experimental analysis. In our experiments, the Pre-CRMSNorm in $\mathbb{R}^{d-1}$ is usually slower than Pre-RMSNorm in $\mathbb{R}^d$, since the accelerators cannot handle $\mathbb{R}^{d-1}$ vectors efficiently when $d$ is a large even number (e.g., 1024).
>
> For inference, we may either (1) add zero paddings, or (2) decompress the vectors and translate the model into the Pre-RMSNorm variant. For training, we suggest keeping the hidden dimension $d$, which is equivalent
> to a Pre-LN model with the hidden dimension $d+1$. In this way, the CRMSNorm can increase model representability and computation efficiency at the same time.
>
> We have trained two groups of models in our early exploration.
>
> 1. Pre-LN in $d = 1024$, Pre-CRMSNorm in $d - 1 = 1023$.
> 2. Pre-LN in $d + 1 = 1025$, Pre-CRMSNorm in $d = 1024$.
>
> The models in each group are equivalent. The accelerators can only handle $d = 1024$ efficiently, with poor performance on $d - 1$ and $d + 1$. To avoid unfair comparison, we only report the results of the first group throughout the paper. For reference, Pre-CRMSNorm in $d$ is 0.9% faster than Pre-LN in $d$.
>
> ### **Slight increase in training workload**
> We provide qualitative analysis in Table 1 and quantitative experimental results in Figure 4a.
> * Table 1. Although recentering and the special linear layer induce extra workload for training, RMSNorm can reduce the training workload significantly such that the total workload is reduced.
> * Figure 4a. The recentering and the special linear induce 1% overhead, while RMSNorm save 3.3% of training time. Overall, the Pre-RMSNorm reduces the training time by 2.3%.
>
> There is no computation overhead for inference since the extra workload can be fused or done before inference.
>
> ### **Decompression in the CRMSNorm**
> Our compression on the zero-mean vectors is lossless, so our decompression can recover the original vectors without loss. We appreciate your attention to this aspect and would be grateful for further details if you can pinpoint any specific concerns.
>
> Thank you so much for reviewing our paper and this response. We are looking forward to further discussions with you.

---

### Author Rebuttal · Authors · 2023-08-10

We sincerely appreciate the constructive feedback provided by all the reviewers. We are truly grateful for their acknowledgment of our paper's contribution and presentation. Engaging with the reviewers is a privilege we highly cherish. Should any queries arise, please do not hesitate to reach out to us. Thank you!

---

### Decision · Program_Chairs · 2023-09-21

**Decision:**

Accept (spotlight)

**Comment:**

This paper proposes a solution to the ongoing debate between Layer Normalization (LayerNorm) and Root Mean Square Normalization (RMSNorm) techniques in Transformers. The authors introduce Pre-LN and Pre-RMSNorm Transformers and demonstrate that by removing redundant mean information, LayerNorm can be reduced to RMSNorm, resulting in higher efficiency without compromising representation ability. Experimental results showcase up to 10% reduction in training and inference time for Pre-LN Transformers. Overall, this paper provides a valuable solution for efficiently normalizing Transformer architectures, offering practical benefits for large language models and other Transformer-based applications. This paper is highly recommended for acceptance as a spotlight paper.